# Investigation of Combinatorial WO_3_-MoO_3_ Mixed Layers by Spectroscopic Ellipsometry Using Different Optical Models

**DOI:** 10.3390/nano12142421

**Published:** 2022-07-14

**Authors:** Miklos Fried, Renato Bogar, Daniel Takacs, Zoltan Labadi, Zsolt Endre Horvath, Zsolt Zolnai

**Affiliations:** 1Institute of Microelectronics and Technology, Kando Kalman Faculty of Electrical Engineering, Óbuda University, H-1084 Budapest, Hungary; bogarrenato@gmail.com (R.B.); tdani98elektro@gmail.com (D.T.); 2Institute of Technical Physics and Materials Science (MFA), Centre for Energy Research, Hungarian Academy of Sciences, H-1525 Budapest, Hungary; labadi.zoltan@energia.mta.hu (Z.L.); horvath.zsolt.endre@energia.mta.hu (Z.E.H.); zolnai.zsolt@energia.mta.hu (Z.Z.)

**Keywords:** spectroscopic ellipsometry, combinatorial approach, metal oxides

## Abstract

Reactive (Ar-O_2_ plasma) magnetron sputtered WO_3_-MoO_3_ (nanometer scaled) mixed layers were investigated and mapped by Spectroscopic Ellipsometry (SE). The W- and Mo-targets were placed separately, and 30 × 30 cm glass substrates were slowly moved under the two (W and Mo) separated targets. We used different (oscillator- and Effective Medium Approximation, EMA-based) optical models to obtain the thickness and composition maps of the sample layer relatively quickly and in a cost-effective and contactless way. In addition, we used Rutherford Backscattering Spectrometry to check the SE results. Herein, we compare the “goodness” of different optical models depending upon the sample preparation conditions, for instance, the speed and cycle number of the substrate motion. Finally, we can choose between appropriate optical models (2-Tauc-Lorentz oscillator model vs. the Bruggeman Effective Medium Approximation, BEMA) depending on the process parameters. If one has more than one “molecular layer” in the “sublayers”, BEMA can be used. If one has an atomic mixture, the multiple oscillator model is better (more precise) for this type of layer structure.

## 1. Introduction

For protection against extra heat through glass windows, electrochromic film as a smart window [1] can be the most useful tool to reduce heat in buildings. A smart glass window consists of a layer of electrochromic material bounded by metal oxide layers. The special feature is the ability to modify the optical properties by supplying electric charge to the film system, which can be transformed from translucent glass into darker or more opaque glass and can be returned to the translucent state with low electric current. It also controls the transmitted amount of light. Electrochromic materials capable of heat radiation protection through glass consist of semiconductor metal oxide film coatings on glass, such as TiO_2_, CrO, Nb_2_O_5_, SnO_2_, NiO, IrO_2_ [2], WO_3_, and MoO_3_ [3,4]. Researchers have different methods of deposition as sputtering [5], Atmospheric Pressure Chemical Vapor Deposition (APCVD) [6], dipping [7], sol-gel method [1,4], and sintering [8]. Authors of Ref. [8] investigated mixed materials, but only a limited number of compositions: (MoO_3_)_x_-(WO_3_)_1__−x_ for x = 0, 0.2, 0.4, 0.6, 0.8. Pure WO_3_ layers were also investigated [9,10,11] by spectroscopic ellipsometry.

During the present work, we used reactive magnetron sputtering (in Ar-O_2_ plasma) to create all combinations of WO_3_-MoO_3_ mixed layers along a line/band. To prepare one sample in the vacuum chamber, we needed 4 h including the vacuum preparation. If we wanted to prepare 21 separate samples with compositions from 0 to 100% with 5% “resolution”, we would need 21 × 4 h, minimum of 10 working days. Using the combinatorial approach, we achieved all of the compositions after one sputtering process in the same sputtering chamber. Furthermore, our aim was to investigate the goodness of WO_3_-MoO_3_ mixed layers as electrochromic materials for “smart” windows (transparency ratio, switching speed, coloration efficiency).

After sputtering, we investigated and mapped the samples by Spectroscopic Ellipsometry (SE), which is a relatively quick, cost-effective, and contactless method. We used different (oscillator- and Effective Medium Approximation, EMA-based) optical models to obtain the thickness and composition map of the sample layer. We checked the SE results using Rutherford Backscattering Spectrometry. In a set of experiments, we changed the position of the sputtering targets, as well as the speed and cycle number of the substrate motion. Our aim was to compare the “goodness” of the different optical models depending upon the sample preparation conditions.

## 2. Materials and Methods

Layer depositions were performed in a reactive (Ar + O_2_) gas mixture in high vacuum (~2 × 10^−^^6^ and ~10^−^^3^ mbar process pressure). Additionally, 30 sccm/s Ar and 30 sccm/s O_2_ volumetric flow rate were applied in the magnetron sputtering chamber. The substrates were 300 × 300 mm soda lime glasses. The starting process was the preparation of a W-mirror (W sputtered only in Ar-plasma) to avoid the back-reflection of the measuring light-beam during Spectroscopic Ellipsometry (SE) measurements. The plasma powers of the two targets were selected in the 0.75–1.5 kW range independently. We used 1, 5 or 25 cm/s of walking speed (back and forth), which was the speed of the 30 × 30 cm glass sample between the end positions (the edges of the targets). See the sample fabrication parameters in Table 1.

Electron Dispersive Spectra (EDS) analysis of the layers showed that the Metal/Oxygen atomic ratio was 1:3 at the applied oxygen partial pressure. Significantly lower oxygen partial pressure is needed to prepare oxygen-deficient (non-transparent, “black”) layer.

The sputtering targets were placed in two arrangements as it can be seen in Figure 1. In the first arrangement, the two targets were placed at 35 cm, in the second arrangement they were placed at 70 cm distance from each other. According to the measurements, in the first arrangement the two “material streams” overlapped around the center position, while in the second arrangement the two “material streams” were separated.

We have two possible optical mapping methods: Our Woollam M2000 SE device [12] or our “expanded beam” ellipsometer [13,14,15]. As a single-spot ellipsometer is more precise when both the thickness and the composition change “rapidly” (in our case ~50 nm and ~10% per cm), we used mainly the M2000 device. In addition, we used CompleteEASE program (from Woollam Co., https://www.jawoollam.com/ellipsometry-software/completeease, accessed 12 July 2022) to evaluate the mapping measurements using the built-in optical models and oscillator functions. Finding the best match between the model and the experiment is typically achieved through regression. An estimator, such as the Mean Squared Error (MSE), is used to quantify the difference between curves. The lower MSE indicates a better fit and better optical model. Notably, the maps from the M2000 measurements are compiled from four 15 × 15 cm parts. Our M2000 device can measure only one 15 × 15 cm part at once. The mapping measurements were performed using mm-sized beam-spot on a 15 × 15 grid with one spectra-pair per cm.

We used 5 × 50 mm Si-probes (6 pieces were placed at the center line of the substrate glass) for Rutherford Backscattering Spectrometry (RBS) and X-ray Diffractometry (XRD) measurements (see later in the “*3.3. Double-Target Samples*”).

Moreover, 2.8 MeV ^4^He^+^ Rutherford Backscattering Spectrometry (RBS) have been performed in a scattering chamber with a two-axis goniometer at 7° tilt and 165° detector angles connected to the 5 MV EG-2R Van de Graaff accelerator of the Wigner FK RMI of the HAS. The ^4^He^+^ analyzing ion beam was collimated with two sets of four-sector slits to the spot size of 0.5 × 0.5 mm (width × height), while the beam divergence was maintained below 0.06°. The beam current was measured by a transmission Faraday cup. In the scattering chamber, the vacuum was about 10^−4^ Pa. Liquid N_2_ cooled traps were used along the beam path and around the wall of the chamber to reduce the hydrocarbon deposition.

RBS spectra were detected using ORTEC silicon surface barrier detectors mounted at scattering angle of *Θ* = 165. The detector resolution was 20 keV for RBS. Spectra were recorded for sample tilt angles of both 7 and 60° to make a difference between the heavier and lighter atoms at the surface and in deeper regions. In this manner, the reliability of spectrum evaluation has been improved. The measured spectra were simulated with the RBX code [16].

XRD measurements were performed on a Bruker AXS D8 Discover device to determine the amount of amorphous fraction of the layers. We examined 4 Si-probes: One from the “W-side”, two from the “mixed-part”, and one from the “Mo-side” and found that our layers are highly amorphous. One example (from the mixed part) XRD measurement is shown in Figure 2. Only one significant broad peak in the 20–30° region can be seen as the sign of amorphous film. Crystalline peaks at higher angles can be identified as peaks of pure cubic (beta) tungsten, which was sputtered under the WO_3_ and MoO_3_ layers. The broad peak near 70° is from the silicon substrate. The vertical red lines show the calculated positions of beta tungsten, which is a thin (app. 100 nm) layer. We calculated the positions of monoclinic, triclinic, and orthorhombic WO_3_ and hexagonal and orthorhombic MoO_3_ peaks. In addition, we cannot see any trace of crystalline WO_3_ or MoO_3_ material in the layers. Moreover, other authors found that independent of the deposition technique, the WO_3_ thin films prepared at room temperature exhibited an amorphous structure (i.e., featureless XRD pattern) [10].

### Dispersion Relations

Magnetron sputtering results in amorphous materials (which is needed for good electrochromic performance) as the XRD measurements prove this (see Figure 2). We considered two different dispersion relations for the clean materials: Cauchy formula (Figure 3a) and Tauc-Lorentz oscillator model (Figure 3b). Both dispersion relations are built-in modules (CompleteEASE program) and can be used as a pre-determined component in Bruggeman Effective Medium Approximation (EMA or BEMA [17]).

Other authors used similar optical models for pure WO_3_ [9,10,11]. In [9], the optical indexes, n and k, were determined by ellipsometric measurements, using various models, including Tauc-Lorentz [18], ensuring a good fit of tan(ψ) and cos(Δ) vs. the wavelength. In [11], the optical constants were measured with the film in the unintercalated and fully intercalated states using ellipsometry, transmission, and reflection data. Two Lorentz oscillators were used to model the dispersion in the 300 and 1700 nm wavelength region. One oscillator in the UV was used to model the dispersion that takes place in dielectric materials for wavelengths larger than the band gap, and a second near 14 μm was used to represent structure in the optical constants due to the tungsten-oxygen network [11]. In the present paper, we do not use the infrared region over 1000 nm, thus we used only one Tauc-Lorentz per material.

In the EMA calculation, the mixed-layer is considered as a physical combination of two distinct phases formed by WO_3_ and MoO_3_ with an appropriate volume fraction. The constituents are considered equivalent; neither of the components is considered as a host material. In this case, it holds that
0 = ∑f*_i_*(ε*_i_* − ε)/(ε*_i_*+2ε),(1)
where ε is the effective complex dielectric function of the composite layer; f_i_ and ε_i_ denote volume fraction and the complex dielectric function of the i^th^ component. In the case of two components, WO_3_ and MoO_3_, the formula is a complex quadratic equation, where ε (the effective dielectric function) is the unknown and we can choose easily between the two solutions (the wrong one is physically meaningless). The used Bruggeman Effective Medium Approximation (EMA or BEMA) is relatively easy to calculate and can be extended simply to describe a material consisting of more than two phases. However, the generalized formula for a two-phase material is
ε = (ε*_a_*ε*_b_*+κε*_h_*(f*_a_*ε*_a_*+f*_b_*ε*_b_*))/(κε*_h_*+(f*_a_*ε*_b_*+f*_b_*ε*_a_*))(2)

Here, κ is defined by κ = (1 − q)/q using the screening factor q. In models that assume spherical dielectrics (i.e., the BEMA model), the screening factor is given by q = 1/3. We tried to use q = 1 (maximal screening) value, as well. However, it provides almost the same results for the compositions and the thickness within the fitting errors, with almost the same, but sometimes a little bit worse fitting quality (higher MSE).

Cauchy formula is good to describe the complex refractive index of low absorption materials: *N = n + ik,* where *N* is the complex refractive index, *n* is the real part of *N, k* is the imaginary part (extinction), *i* is the imaginary unit: *n(**λ) =*
*A+B/**λ^2^+**C/**λ^4^;*
*k(**λ) =*
*U_1_e ^U^_2_^( 1239.84/λ^
**^−^**^Eb)^*, where A, B, C, U_1_, and U_2_ are fitting parameters. The complex refractive index *(N)* and complex dielectric function (*ε*
*=*
*ε_1_+iε_2_*) are equivalents: *ε*
*= N**^2^;*
*ε_1_ = n**^2^** − k**^2^,*
*ε_2_ = 2nk*. The main drawback of the Cauchy formula is that it is good only below the bandgap.

The Tauc-Lorentz (T-L) oscillator model is a combination of the Tauc and Lorentz models [18]. T-L model contains four parameters: Transition Amplitude, Broadening coefficient of the Lorentz oscillator, peak position for the Lorentz oscillator, and Bandgap Energy (E_g_), which is taken to be the photon energy, where ε_2_ (E) reaches zero. When the E photon energy is less than the bandgap energy, Eg, ε_2_ (E) is zero. The real part of the dielectric function ε_1_ (E) can be obtained from ε_2_ (E) through the Kramers-Kronig relation.

We determined the dispersions for pure (100%) WO_3_ and MoO_3_ using both the Cauchy formula and the Tauc-Lorenz model. Considering the fitted SE spectra (for example, 100% WO_3_ in Figure 4) using the Cauchy and Tauc-Lorentz (T-L) formulas, we can see, especially in the UV part, that the Tauc-Lorentz oscillator model is better (lower MSE) for these materials, even below 300 nm. Notably, the T-L formula has only 4 parameters. During the following optical models, we used the determined complex refractive indices (and complex dielectric functions) of the pure WO_3_ and MoO_3_ using the Tauc-Lorentz oscillator model, see Figure 3b. Finding the best match (see Figure 4) between the model and the experiment is typically achieved through regression. An estimator, such as the Mean Squared Error (MSE), is used to quantify the difference between curves.

## 3. Results and Discussion

### 3.1. Single-Target Samples

First, we performed single target depositions to assess the lateral distribution of sputtered material flux in the case of single W- and single Mo-targets at different electrical powers. When the composition is not changing, the expanded beam mapping is not bad, see Figure 5. Herein, we compare the results of expanded beam mapping (Figure 5a) and the Woollam M2000 map (Figure 5b). Notably, the M2000 map is compiled from four independently measured 15 × 15 cm parts and we tried to compose the whole map along the “iso-thickness” or “iso-color lines”.

One can find a good summary regarding the “Mapping and imaging of thin films on large surfaces” in [19].

### 3.2. Thickness vs. Power

We performed single-target experiments to assess the deposition rate dependence on the power and on the distance from the target. In this way, we determined the angle dependence of “the material stream”. We used 1.5, 1, and 0.75 kW powers. The layer deposition rate is non-linear: Double power results in 7 times higher rate in the case of WO_3_. Figure 6, Figure 7 and Figure 8 are not fully relevant for the “double-target” experiments due to the different “walking” speeds and distances. “W-target only” and “Mo-target only” indicate that only the target was under electrical power during the deposition process. “Walking” speed indicates the speed of the 30 × 30 cm glass sample between the end positions (the edges of the targets). All these maps were measured by our Woollam M2000 SE device.

Similar measurements were performed for the Mo-target, as well. We show here only the 1 kW case, since it provides similar results to the “W-target only” case. Additionally, we used the small Si-probes (at the center lines of the 30 × 30 cm glass sheets, see later in the “*3.3. Double-Target Samples*”) to determine the thickness by RBS as an independent method.

### 3.3. Double-Target Samples

#### 3.3.1. Targets in Closer Position

The first (combinatorial) experiment was performed in the “targets in closer position” (Figure 1a); the power of the W-target was 0.75 kW and the power of the Mo-target was 1.5 kW. The choice of the W/Mo power ratio was based on the individual thickness profiles and was created to ensure that the 50% composition falls in the middle of the sample. Additionally, 300 walking cycles were applied with 5 cm/s walking speed. (We can calculate ~1 nm of sublayer thickness around the center part, where the 50% mixture is expected).

We used a 2-Tauc-Lorentz (2-T-L) oscillator optical model: W-substrate/interface-layer/T-L(WO_3_)+T-L(MoO_3_)-mixed-layer/surface-roughness-layer. (This model layer is better for atomic mixture). Additionally, five fitted parameters were used: Layer thickness and the two Amplitudes (oscillator strengths). The basic parameters (the Broadenings, the Peak positions, and the Bandgap Energies) of the clean materials were determined from the measurements near the edges of the samples. The typical values of surface roughness and interface thickness are not more than 10 and 20 nm, see, for example, Figure 9b.

Figure 9 shows the resulted maps (with the schematic optical model): Thickness-map and two Amplitude-maps (T-L(WO_3_)–map and T-L(MoO_3_)–map). One can see that the composition (Amplitudes) changes from 0 to 100% in the middle of 10–15 cm wide range. 

The Amp_WO3_/Amp_WO3-100%_ and Amp_MoO3_/Amp _MoO3-100%_ ratios “move” to the opposite direction (Amp_WO3_ = Amp1 and Amp_MoO3_ = Amp2 the fitted parameters, see Figure 9b–d) and these Amplitude ratios are good estimators (within the fitted errors) for the W/Mo atomic ratio. We validated the results with the RBS results shown in Figure 10.

The composition at the different lateral positions were checked by RBS measurements, as well (see Figure 10 and Figure 11), which shows a good agreement between the results of the two methods. Notably, the yield counts of the background free oxygen signal (channels #200–320) in the RBS spectra provides an uncertainty less than 8% for the stoichiometric ratio of oxygen in the (W_x_Mo_y_)O_3_ layer, while for Mo and W, the error of x and y is less than 2% due to their significantly higher RBS yields. Nevertheless, the three components are fitted together, thus their atomic ratios are coupled in the simulation when looking for the best fit of measured data (red line). This provides a lower limit for the error of the oxygen content, as well. Therefore, the highest error 2% (±0.06) can be considered for the stoichiometric index of O.

#### 3.3.2. Atomic (or “Molecular”) Mixture vs. “Superlattice”

We prepared two different samples in the “targets at distant position” mode (Figure 1b). The difference was the walking speed (and the number of the walking cycles): One “Fast” (walking speed: 25 cm/s) and one “Slow” (walking speed: 1 cm/s) sample. The different speeds resulted in different “sublayer thickness” of ~0.5 nm for the “Fast” sample and 3–5 nm for the “Slow” sample, calculated from the final thickness and the number of the walking cycles around the center part, where the 50–50% mixture is expected. We used two types of optical models: 2-T-L oscillator model and Effective Medium Approximation (EMA [17]) model (see Figure 9, upper right). Finding the best match between the model and the experiment is typically achieved through regression. An estimator, such as the Mean Squared Error (MSE), is used to quantify the difference between curves. We can choose between the optical models with the MSE-maps: The model is better if the MSE values are significantly lower at the relevant (around 50–50% composition) positions (see for example Figure 12). These figures show only the interesting quarters of the maps where the composition is changing faster.

##### “Fast” Sample, 25 cm/s Walking: ~0.5 nm “Sublayer Thickness”

The thickness of the layer around the center part is app. 1000 nm. We can calculate a “sublayer thickness” of ~0.5 nm for this “Fast” sample from the number of the walking cycles (1500) and we can consider it an atomic mixture. Thickness-maps (Figure 13) show nearly the same results for both optical models. The Amplitude-of-T-L (only WO_3_)–map and the EMA (volume percent of WO_3_)–map (Figure 14) and MSE-maps (Figure 12) show similar tendencies. However, the 2-T-L oscillator model shows significantly lower MSE values against the Effective Medium Approximation model, see the red ellipses in Figure 12. The significantly lower MSE values (around the 50–50% ratio) show that the 2-T-L oscillator model is better for this type (atomic mixture) of layer structure.

The measured and fitted example spectra in Figure 15 show that the 2-T-L model is better, especially under the 450 nm wavelength region (see red ellipses) where the light absorption is significant. 

##### “Slow” Sample, 1 cm/s Walking: 3–5 nm Sublayer Thickness

The thickness of the layer around the center part is app. 280 nm. We can calculate a “sublayer thickness” of ~4 nm for this “Slow” sample from the number of the walking cycles (70) and we can consider it as a “superlattice”. (We can call it is superlattice only at 50–50%, otherwise we could call it a type of superlattice).

The Amplitude-of-T-L (only WO_3_)–map, MSE-maps (Figure 16), thickness-maps (Figure 17) and the EMA (volume percent of MoO_3_)–map (Figure 18) show similar tendencies. However, the Effective Medium Approximation model shows significantly lower MSE values against the 2-T-L-oscillator-model, see the red ellipses in Figure 16. The significantly lower values around the center part, especially around the 50–50% ratio, show that the Effective Medium Approximation oscillator model is better for this “superlattice” type of layer structure. The measured and fitted example spectra in Figure 18 show that the Effective Medium Approximation model is better with a 1.3% precision of the composition (volume fraction) parameter.

The measured and fitted example spectra in Figure 19 show that the EMA model is better for the “superlattice” type of layer structure.

## 4. Conclusions

In summary, we can produce combinatorial samples on large scale in a magnetron sputtering system. These samples can be mapped (thickness and composition maps, as well) in a fast and non-destructive manner by Spectroscopic Ellipsometry. Moreover, we can choose between appropriate optical models (2-Tauc-Lorentz oscillator model vs. the Bruggeman Effective Medium Approximation, BEMA) depending on the process parameters. In conclusion, if one has more than one “molecular layer” in the “sublayers”, BEMA can be used. If one has an atomic mixture, the multiple oscillator model is better (more precise) for this type of layer structure.Moreover, our conclusion is that in the case of “atomic mixing”, the two Amplitudes of the 2-Tauc-Lorentz oscillator (one T-L oscillator for the WO_3_ component and one T-L oscillator for the MoO_3_ component) is a better estimator for the atomic ratio of the W/Mo ratio.

In this way, we have a fast and non-destructive (contactless) method to determine the position dependent composition of our combinatorial samples when we measure the optimal electrochromic behavior of these samples [20].

## Figures and Tables

**Figure 1 nanomaterials-12-02421-f001:**
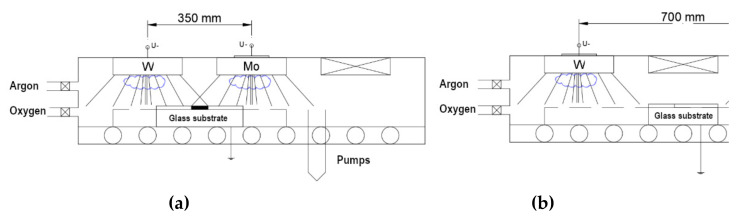
Two arrangements of the targets: (**a**) The two targets in closer position (35 cm from each other); (**b**) the two targets in distant position (70 cm from each other).

**Figure 2 nanomaterials-12-02421-f002:**
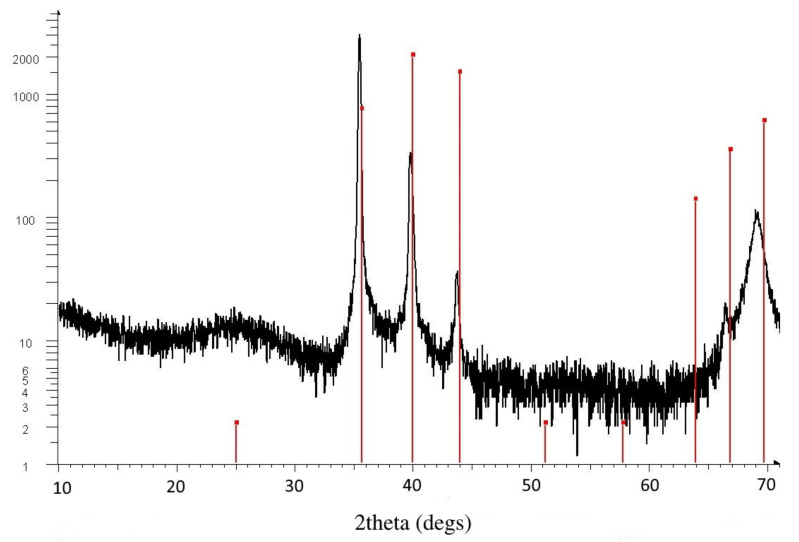
One example (from the mixed part) XRD measurement (note the logarithmic vertical scale) showing one significant broad peak in the 20-30° region as the sign of amorphous film. Crystalline peaks at higher angles can be identified as peaks of pure cubic (beta) tungsten, which was sputtered under the WO_3_ and MoO_3_ layers. In addition, the broad peak near 70° is from the silicon substrate. The vertical red lines show the calculated positions of beta tungsten. We calculated the positions of monoclinic, triclinic, and orthorhombic WO_3_ and hexagonal and orthorhombic MoO_3_ peaks. Moreover, we cannot see any trace of crystalline WO_3_ or MoO_3_ material in the layers.

**Figure 3 nanomaterials-12-02421-f003:**
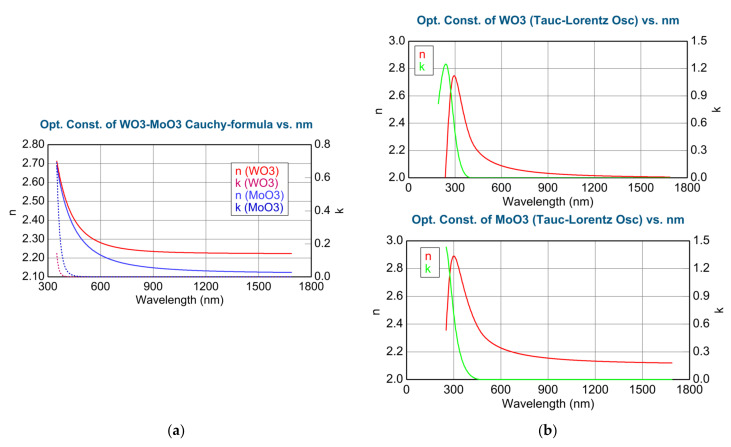
Determined complex refractive indices: (**a**) Using the Cauchy formula for the pure WO_3_ and MoO_3_; (**b**) using the Tauc-Lorentz oscillator model for pure WO_3_ (upper) and MoO_3_ (lower).

**Figure 4 nanomaterials-12-02421-f004:**
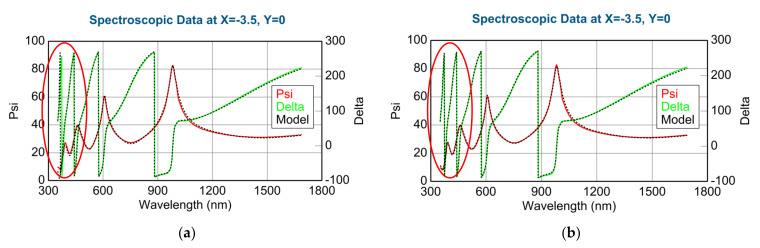
Measured (continuous lines) and fitted (dotted lines) SE spectra for the 100% WO_3_ layer composition using Cauchy (**a**) and Tauc-Lorentz (T-L) (**b**) formulas. Red ellipses show the region where the quality of the fit is different, and is better using the Tauc-Lorentz formula.

**Figure 5 nanomaterials-12-02421-f005:**
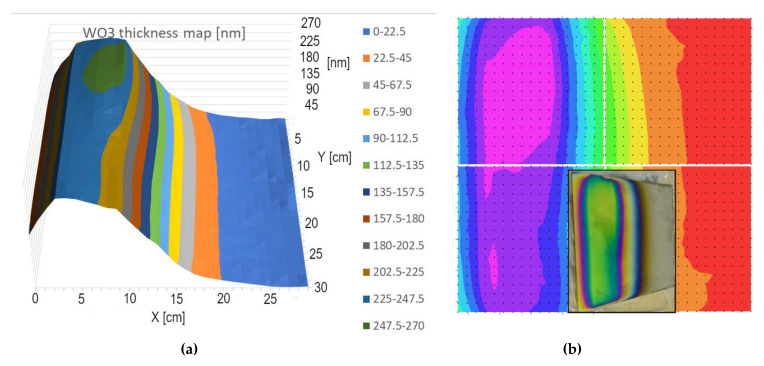
Thickness-maps (in nm) of a 30 × 30 cm sample. Maximum thickness value is 270 nm. One can compare (**a**) the results of expanded beam mapping and (**b**) the Woollam M2000 map. Notably, the M2000 map is compiled from four independently measured 15 × 15 cm parts. The photograph (inserted) shows the direct view of the sample.

**Figure 6 nanomaterials-12-02421-f006:**
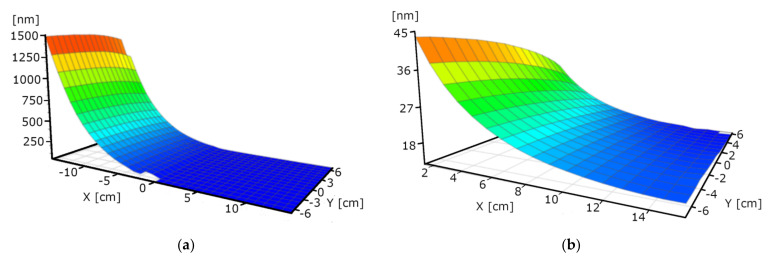
Thickness-maps (in nm) of “W-target only” sample at 1.5 kW power. Maximum thickness value is around 1500 nm. The left map (**a**) shows the middle 15 × 30 cm part, the right map (**b**) shows the thin part with magnified scale (maximum thickness value is around 45 nm).

**Figure 7 nanomaterials-12-02421-f007:**
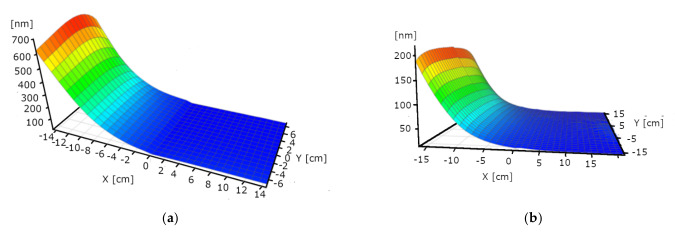
Thickness-maps (in nm) of “W-target only” samples at 1 (**a**) and 0.75 (**b**) kW power. Maximum thickness value is around 700 and 200 nm, respectively.

**Figure 8 nanomaterials-12-02421-f008:**
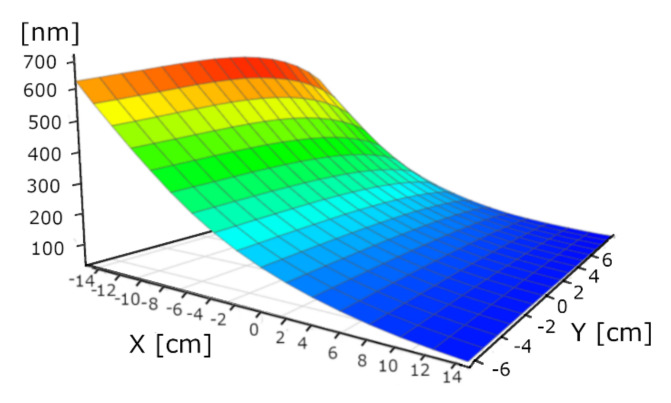
Thickness-map (in nm) of “Mo-target only” sample at 1 kW power. Maximum thickness value is around 700 nm, which corresponds to the W maximum thickness.

**Figure 9 nanomaterials-12-02421-f009:**
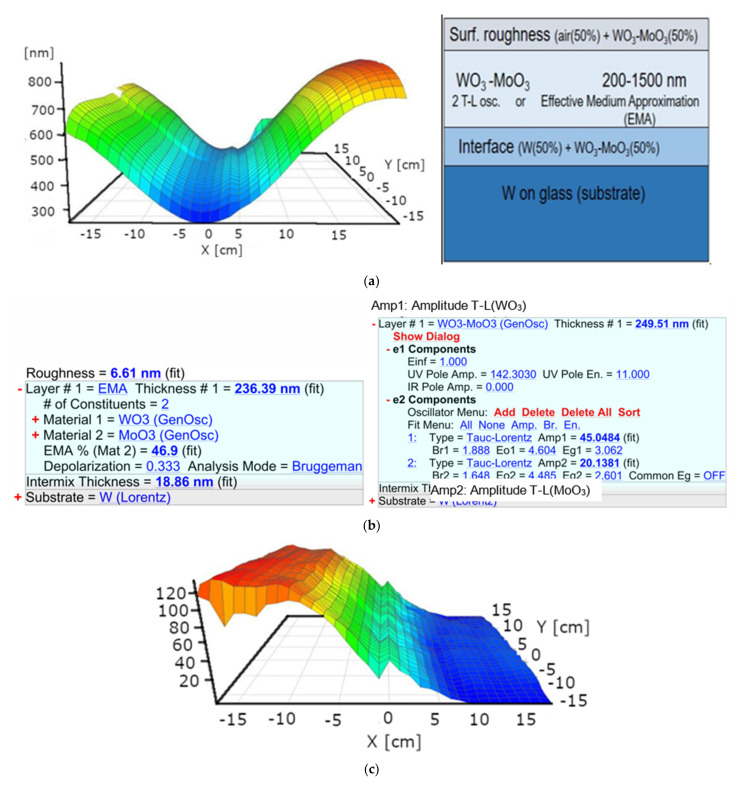
(**a**) Thickness-map and the schematic optical model; (**b**) two versions of the optical model: The Bruggeman Effective Medium Model (left), the 2-Tauc-Lorentz (2-T-L) oscillator model (right), “(fit)” show the fitted parameters; (**c**) Amp1: Amplitude-of-T-L(WO_3_)–map; (**d**) Amp2: Amplitude-of-T-L(MoO_3_)–map (the wrinkles at the center lines are artefacts caused by the manual “rotation” during the SE measurement).

**Figure 10 nanomaterials-12-02421-f010:**
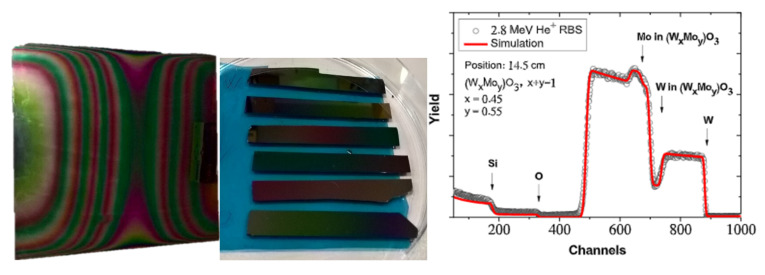
Photograph of one sample (left). Photograph of the Si-probes, which were placed at the center line of some samples for RBS and XRD (center). One Rutherford Backscattering Spectrometry example near the center position.

**Figure 11 nanomaterials-12-02421-f011:**
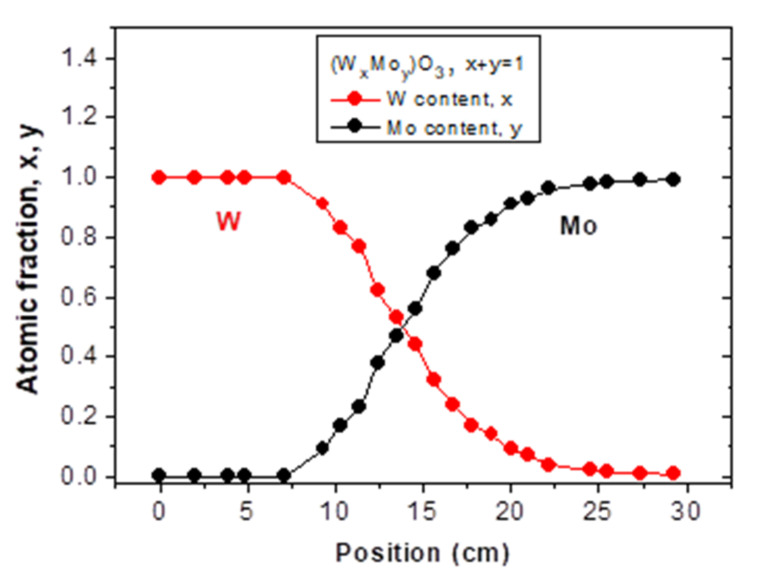
Composition-map along a central line by Rutherford Backscattering Spectrometry.

**Figure 12 nanomaterials-12-02421-f012:**
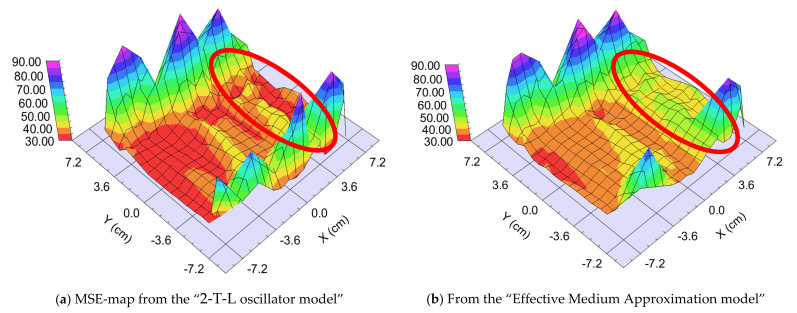
Mean Squared Error (MSE)-maps (**a**) using the 2-Tauc-Lorentz (2-T-L) oscillator model (**b**) and the Effective Medium Approximation model. Red ellipses show the interesting area, where the composition changes the most. We show only one 15 × 15 cm part, all other parts show the same tendencies.

**Figure 13 nanomaterials-12-02421-f013:**
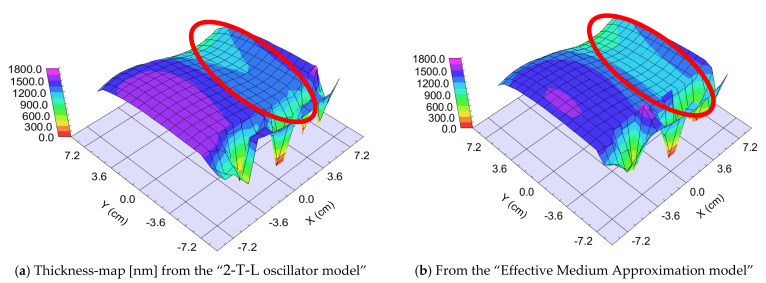
Thickness-maps (**a**) using the 2-Tauc-Lorentz (2-T-L) oscillator model (**b**) and the Effective Medium Approximation model. Red ellipses show the interesting area, where the composition changes the most. We show only one 15 × 15 cm part, all other parts show the same tendencies.

**Figure 14 nanomaterials-12-02421-f014:**
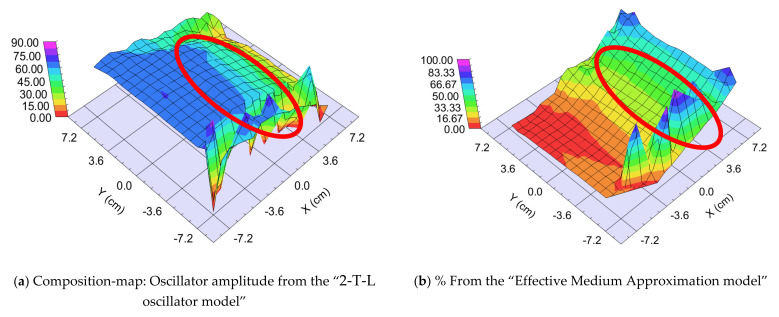
Amplitude-of-T-L (only WO_3_)–map (**a**) and EMA% (MoO_3_)–map (**b**). Red ellipses show the interesting area, where the composition changes the most. We show only one 15 × 15 cm part, all other parts show the same tendencies.

**Figure 15 nanomaterials-12-02421-f015:**
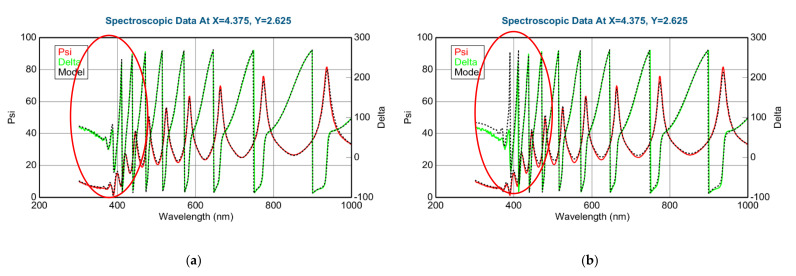
Measured and fitted spectra at one sample point: 2-Tauc-Lorentz (2-T-L) oscillator model. (**a**) Fit error (MSE) = 30.4, Thickness = 1159.6 ± 2.9 nm, Amp1 = 39.2 ± 1.1, Amp2 = 26.7 ± 0.4), and Effective Medium Approximation model; (**b**) fit error (MSE) = 40.1, Thickness = 1081.7 ± 2.8 nm, EMA% (Mat2) = 41.8 ± 0.8).

**Figure 16 nanomaterials-12-02421-f016:**
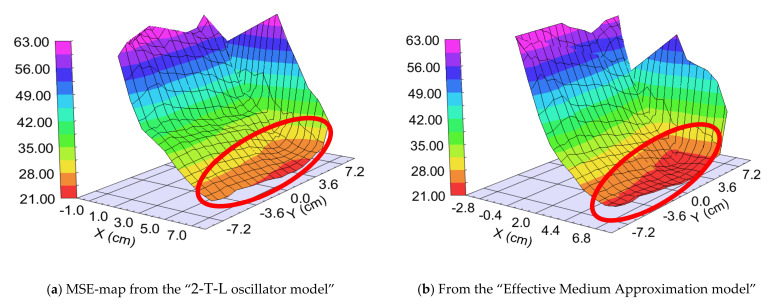
Mean Squared Error (MSE)-maps (**a**) using the2-Tauc-Lorentz (2-T-L) oscillator model (**b**) and the Effective Medium Approximation model. Red ellipses show the interesting area, where the composition changes the most. We show only one 15 × 15 cm part, all other parts show the same tendencies.

**Figure 17 nanomaterials-12-02421-f017:**
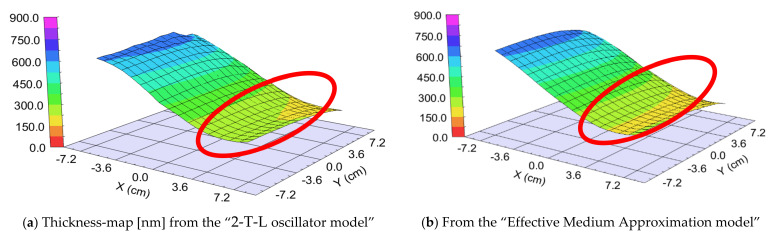
Thickness-maps (**a**) using the 2-Tauc-Lorentz (2-T-L) oscillator model (**b**) and the Effective Medium Approximation model. Red ellipses show the interesting area, where the composition changes the most. We show only one 15 × 15 cm part, all other parts show the same tendencies.

**Figure 18 nanomaterials-12-02421-f018:**
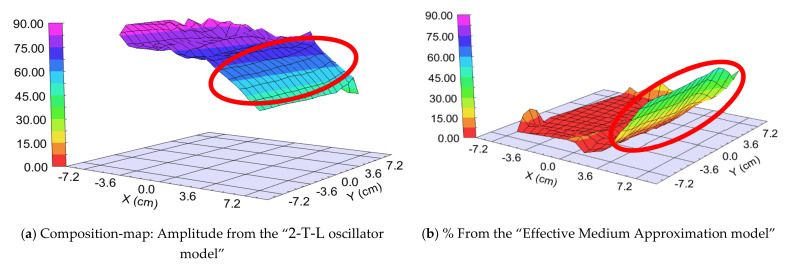
Amplitude-of-T-L (only WO_3_)–map (**a**) and EMA% (MoO_3_)–map (**b**). Red ellipses show the interesting area, where the composition changes the most. We show only one 15 × 15 cm part, all other parts show the same tendencies.

**Figure 19 nanomaterials-12-02421-f019:**
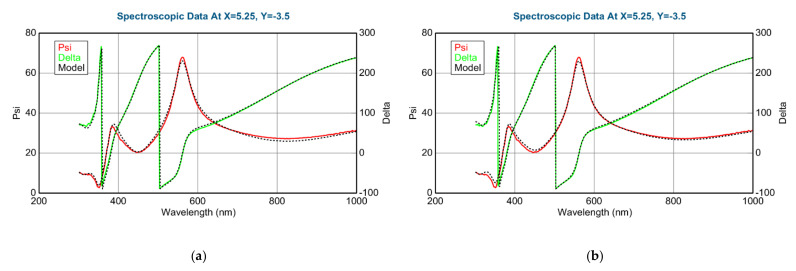
Measured and fitted spectra at one sample point: 2-Tauc-Lorentz (2-T-L) oscillator model. (**a**) Fit error (MSE) = 29.2, Thickness = 230.4 ± 0.8 nm, Amp1 = 58.0 ± 1.5, Amp2 = 11.4 ± 0.7, and Effective Medium Approximation model; (**b**) fit error (MSE) = 24.8, Thickness = 218.0 ± 0.1 nm, EMA% (Mat2) = 28.5 ± 1.3.

**Table 1 nanomaterials-12-02421-t001:** Summary of the sample fabrication conditions.

Sample Name	Target (s)	Target Position	Plasma Powers [kW]	Walking Cycles	Walking Speed
W-target-only	W	center	0.75	500	5 cm/s
W-target-only	W	center	1	500	5 cm/s
W-target-only	W	center	1.5	500	5 cm/s
Mo-target-only	Mo	center	0.75	500	5 cm/s
Mo-target-only	Mo	center	1	500	5 cm/s
Mo-target-only	Mo	center	1.5	500	5 cm/s
Double-target in closer position	W-Mo	Left-center	0.75–1.5	300	5 cm/s
Double-target in distant position “Slow”	W-Mo	Left-right	0.75–1.5	75	1 cm/s
Double-target in distant position “Fast”	W-Mo	Left-right	0.75–1.5	1500	25 cm/s

## Data Availability

Not applicable.

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
