# Peer review of "Investigation of Combinatorial WO3-MoO3 Mixed Layers by Spectroscopic Ellipsometry Using Different Optical Models"

_nanomaterials, 2022, doi:10.3390/nano12142421_

Round 1

Reviewer 1 Report

A review of your article can be found in the attached file.

Author Response

I uploaded the Answers to Ref-1

Reviewer 2 Report

In this work the authors propose a particular method to have a continuously graded WO3+MoO3 electrochromic layer on metal tungsten/SLG substrate. The film thickness of films produced by different power of magnetron sources and different sliding speeds of the substrate is proposed in order to ontain a region with continuous intermixing of two amorphous WO3 and MoO3 oxides.

Spectroscopic ellipsometry mapping as rutherford back-scattering as well is used to map the gradients in composition of two co-sputtered oxides.

The main problem I have found and that the authors show in fig.9 and fig.18 is that the amplitude of the oscillators related to the Tauc-Lorentz oscillators, that fits pretty good for the investigated 300-1700nm wavelength range for transparent amorphous oxide with a particular band-gap, is a property of material itself, and should not change in response to the position.

One expects that eventually only thickness (or equivalent) or volume fraction may change versus position in the maps.

A second but a bit less important point is about the choice of the depolarization factor in the EMA formulation: the formula 1 at page 5 depolarization is by design set to 1/3 and this suppose that one have spherical inclusions in the host matrix, which maybe the case (or not). Linear combination of the dielectric constants imply depolarization=0 and so a 'columnar like' juxtaposed materials, (H. Fujiwara 'Spectroscopic ellipsometry. principles and applications' ISBN 4 621 07253 6 pagg.179-180) which seems not really fit to describe the nearly molecular level intermixing of the two oxides for 'fast sliding' samples.

Figure 2: the two upper inset are unreadable

Figure 3-4: import better quality images, the graphs are very bad... or import the data into a scientific graphing software like origin or igor: you only need to go with mouse cursor on the graph of CompletEase and press ctrl+c to copy all data in text format on the clipboard, and them you can just save a csv ot column-arranged text file to be processed.

Figure 5b: there are composition issues in the pdf I have

Figure 6-8: Z axis scale must be better sorted 

All these main notes plus other minors issues are annotated in the reviewed  pdf file in attachment

Best regards  

Author Response

I uploaded the Answers to Ref-2

Round 2

Author Response

Answer to Reviewer2:

Many thanks for the opinion and suggestions. We enhanced and changed the Fig. 1, 2 and 5. We hope that no more problems remained in the manuscript.

This manuscript is a resubmission of an earlier submission. The following is a list of the peer review reports and author responses from that submission.

Round 1

Reviewer 1 Report

The paper should be accepted after major revisions. An overall comment is that the figures are not enough described in the text. The use of the combinatorial chemistry using two different arrangements for the sputtering set up is of interest for the community.

Additional references have to be added for discussion on ellipsometry as well as refractive index determination, see for instance ;

Morphological and chemical dynamics upon electrochemical cyclic sodiation of electrochromic tungsten oxide coatings extracted by in situ ellipsometry, Zimmer, A., et al., Applied Optics, 2020, 59(12), pp. 3766–3772

Electrochromic WO3 thin films active in the IR region, Sauvet, K., et al. Solar Energy Materials and Solar Cells, 2008, 92(2), pp. 209–215

The sentence : XRD measurements…to examine the low crystallinity..is rather strange..and needs to be changed. Or at least the part to examine the low crystallinity needs to be deleted.

Figure 2 is not sufficiently described in the text. This part needs to be clarified.

Figure 5 : The thickness mapping is a relevant study useful for the community. Can the authors compare their results with existing literature ?

Figure 8 : Why only 1 kW for the Mo target ? and not 1 and 0.75 kW as shown for W target only?

The choice of power for the double-target samples need to be better explained ?

Regarding the film composition, any comments on the oxygen stoichiometry ? always 3 ? Do the authors have any proof ?

Minor comment : Dots need to be used in Figure 11, 0,2 to be replaced by 0.2

Reviewer 2 Report

The manuscript describes exploitation of several methods for determination of thin layer properties. Honestly, I am still trying to find the novelty and the idea of the research. The description of the fabrication technology is very trivial and the intention of the result is not very clean so I have only a rough idea about produced samples. There is quite a chaos in the number of samples produced at different conditions and I missed the information about number of sweeps of the substrates in the sputtering process under the two sources. A table of investigated samples and their fabrication conditions would be helpful. The manuscript consist of inappropriate number of graphs without detailed description. Figure 5b is scattered and cannot be observed. I could not find the information about methods used for obtaining the thickness maps in Figures 6-8, and I can only estimate that it was the Woollam M2000 SE device. Was the thickness of the layers measured by other method to prove the results from SE measurements, for example by mechanical stylus profilometer? There is lots of other information missing. For example, what are the MSE error maps on figures 12 and 16 related to, is it the RBS measurement? Which part of the samples are depicted in Figures 12-14 and 16-18, as the samples we 30x30 cm2 and the pictures are in smaller areas? The interesting part of the research is the determination of composition using SE method, but the results are not properly verifiable as the results from SE are not possible to compare with RBS measurements. And the main question is the scientific novelty of the research? Well known analytical methods are used for investigation of badly described combinatorial layers using well known technology is not really a story for journal with IF 5.076 and I need to reject the paper in current form. You may try to convince me if you rebuild the manuscript to show the clear research intention and usefulness of it, but in current form it is not publishable.

Reviewer 3 Report

The authors have investigated electrochromic properties of WxMo1-xO3, and are going to publish the results in Applied Surface Science [16]. This is perfectly OK.

Furthermore, the authors apparently tried to increase the number of their publications by writing a manuscript for Nanomaterials which describes how they obtained part of these data (especially elemental composition as indicated by the optical properties), including stuff such as performance of not only the optimum optical model but also some suboptimum optical models. This is not OK. I cannot find any reason why to publish the present manuscript. I do not see it attracting many citations. The manuscript looks like a student's report for his supervisor ... most of the figures included should constitute a basis for internal discussions in the author's lab rather than a standalone paper. The recommended optical model - Tauc-Lorentz oscillator & effective medium approximation - is correct but pretty standard and does not constitute anything novel. Furthermore, the manuscript is about effectively homogeneous amorphous film, i.e. it has very little to do with Nanomaterials or Nanotechnology.

I could put technical comments, ranging from too small text size is many figures through not very consistent considering the Urbach tail in the Cauchy case but not in the Tauc-Lorentz case (which would lead to Cody-Lorentz) to the fact that the WO3 and MoO3 properties (refractive index, optical band gap which is a parameter of the Tauc-Lorentz oscillator) are not compared with the literature in order to confirm film quality. However, it is not necessary in this particular case ... I do not think that the aforementioned main concern can be changed by a textual revision.